# Quantum Atomic Arrays: Fractional Filling and Trapping

Pengfei Zhang

Institute for Quantum Information and Matter and Walter Burke Institute for Theoretical Physics,
California Institute of Technology, Pasadena, CA 91125, USA
* PengfeiZhang.physics@gmail.com

August 26, 2021

## Abstract

**Quantum emitters, in particular, atomic arrays with subwavelength lattice constant, have been proposed to be an ideal platform for study the interplay between photons and electric dipoles. In this work, motivated by the recent experiment [1], we develop a microscopic quantum treatment using annihilation and creation operator of atoms in deep optical lattices. Using a diagrammatic approach on the Keldysh contour, we derive the cooperative scattering of the light and obtain the general formula for the $S$ matrix. We apply our formulism to study two effects beyond previous treatment with spin operators, the effect of fractional filling and trapping mismatch. Both effects can lead to the imperfectness of atomic mirrors. For the fractional filling case, we find the cooperative linewidth is linear in filling fraction $n$. When there is a mismatch between the trapping potentials for atoms in the ground state and the excited state, multiple resonances can appear in the response function. Our results are consistent with existing experiments.**

## 1 Introduction

The ability to coherently storing photons and controlling their interaction with quantum matters is of vital importance for quantum science. Although single atoms and photons usually interact

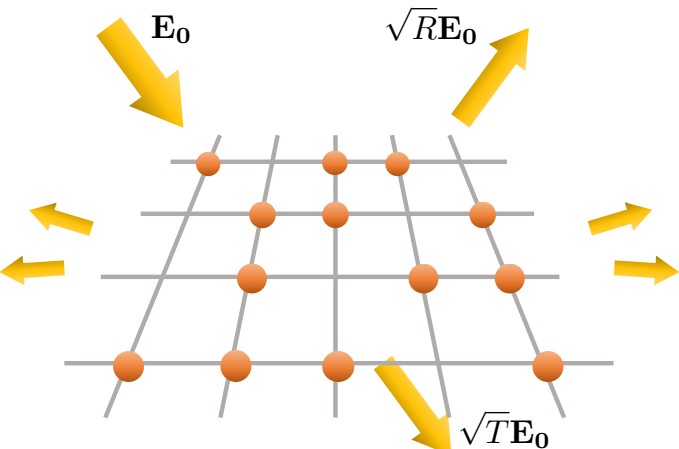

Figure 1: Schematics of the model considered in this work: the atomic array in the optical lattices at fractional filling.

less efficiently, ensembles of atoms can show a cooperative response of photons. As an example, superradiance can be realized when the radiations between atoms interfere constructively [2–8]. Recently, atomic arrays with subwavelength lattice structures are found to be an ideal platform where electric dipole-dipole interactions between atoms are mediated by photons [9–18]. The analysis shows the atomic arrays exhibit subradiance and are nearly perfect mirrors for a wide range of incident angles [16], as observed in recent experiments [1]. Later, there are many theoretical studies on the fruitful physics in atomic arrays [19–26, 26–37]. For example, there are proposals for realizing non-trivial topology in atomic arrays [19–21], controlling atom-photon interaction using atomic arrays [22–26], and efforts in understanding their subradiant behaviors and ability of photon storage [26–32].

In most of these works, atoms are treated as point-like with no motional degree of freedom. The evolution of the system is described by using non-Hermitian Hamiltonian or Lindblad master equation [16, 17], with spin degree of freedom $\sigma_{im}^{-} = |\mathbf{r}_m, g\rangle\langle\mathbf{r}_m, e_i|$. Here $|\mathbf{r}_m, g\rangle$ is the $s$-wave ground state for the atom at position $\mathbf{r}_m$. $|\mathbf{r}_m, e_i\rangle$ is the $p$-wave excited labeled by the dipole moment $\mathbf{d} = d\,\mathbf{e}_i$ of the corresponding transition $g \rightarrow e_i$. However, in real experiments, the system consists of atoms moving in optical lattices. For deep optical lattices, although atoms are trapped near the potential minimum, the wave function for the motional degree of freedom may still play a role. Moreover, the consequence of fractional filling has been studied in the experiment. It is difficult to analyze the absence of an atom in the spin-operator language, and consequently, theoretical predictions for the fractional filling case are still absent.

In this work, we overcome this difficulty by using a microscopic model for the coupled system consisting of atoms in deep optical lattices and photons. After making plausible assumptions, we derive the cooperative response of the system using a diagrammatic approach on the Keldysh contour. By summing up bubble diagrams with dressed Green's function, we obtain neat results for the transmission coefficient and the reflection coefficient, with the contribution from motional wave function. Our result matches the previous analysis for unit filling when the potential of the excited state atoms are the same as that of the ground state. For fractional fillings, we find the cooperative linewidth is linear in $n$, consistent with the experimental observation. We then study the effect of the discrepancy of optical lattices for the ground state and excited state atoms, where the transition of the internal state can be accompanied by transitions in the motional degree of freedom. In particular, we find that multiple resonances can exist in the response function.

## 2 Model

We consider coupled systems with atoms and photons. Th Hamiltonian reads

$$H = H_{\text{EM}} + H_{\text{A}} + H_{\text{int}}. \tag{1}$$

Here the first term is the Hamiltonian of the electromagnetic field

$$H_{\text{EM}} = \int d\mathbf{r} \left( \frac{\epsilon_0}{2} \mathbf{E}(\mathbf{r})^2 - \frac{\mu_0}{2} \mathbf{H}(\mathbf{r})^2 \right), \tag{2}$$

The second term describes the motion of atoms in optical lattices

$$H_{\text{A}} = \int d\mathbf{r} \sum_i \psi_{e_i}^\dagger(\mathbf{r}) \left( \omega_0 - \frac{\nabla^2}{2} + V_e(\mathbf{r}) \right) \psi_{e_i}(\mathbf{r}) + \int d\mathbf{r} \, \psi_g^\dagger(\mathbf{r}) \left( -\frac{\nabla^2}{2} + V_g(\mathbf{r}) \right) \psi_g(\mathbf{r}), \tag{3}$$

$V_{g/e}(\mathbf{r})$ describes the optical lattice potential for ground/excited-state atoms. We have set $\hbar = 1$ and $m = 1$. We assume each site is occupied by at most one atom, which corresponds to choosing fermionic commutation relation $\{\psi_a^\dagger(\mathbf{r}), \psi_a(\mathbf{r}')\} = \delta_{ab}\delta(\mathbf{r} - \mathbf{r}')$. The last term describes the interaction between atoms

$$H_{\text{int}} = -\int d\mathbf{r} \sum_i \left( P_i^+(\mathbf{r}) + P_i^-(\mathbf{r}) \right) \mathbf{e}_i \cdot \mathbf{E}(\mathbf{r}), \tag{4}$$

with

$$P_i^+(\mathbf{r}) = d \, \psi_{e_i}^\dagger(\mathbf{r})\psi_g(\mathbf{r}) = P_i^-(\mathbf{r})^\dagger. \tag{5}$$

The full Hamiltonian (1) describes general model with interaction between atoms and light to the order of electric dipole transition. For atomic arrays, the ground state particle is always tightly trapped near the local minimum of optical lattices, with a spread of wave function $\sigma \ll a$, where $a$ is the lattice constant [1]. Assuming the excited state is also trapped, we expand

$$\psi_g(\mathbf{r}) \approx \sum_n \varphi_a(\mathbf{r} - \mathbf{r}_n)\psi_g^a(\mathbf{r}_n), \qquad \psi_{e_i}(\mathbf{r}) \approx \sum_n \varphi_a'(\mathbf{r} - \mathbf{r}_n)\psi_{e_i}^a(\mathbf{r}_n). \tag{6}$$

Here $\varphi_a(\mathbf{r})/\varphi_a'(\mathbf{r})$ is the motional wave function for ground/excited-state atoms near the local minimum $\mathbf{r}_n = \mathbf{0}$ with the energy $e_a/e_a'$, and we have $\mathbf{r}_n = a(n_1\mathbf{e}_1 + n_2\mathbf{e}_2)$. The commutation relation for $\psi_\eta^a(\mathbf{r}_n)$ now becomes $\{\psi_\eta^{a,\dagger}(\mathbf{r}_m), \psi_\xi^b(\mathbf{r}_n)\} = \delta_{\eta\xi}\delta_{ab}\delta_{mn}$. Using (6), the Hamiltonian $H_{\text{A}}$ and $H_{\text{int}}$ can be simplified. We have

$$H_{\text{A}} = \sum_{na} \left[ \sum_i e_a' \psi_{e_i}^{a,\dagger} \psi_{e_i}^a(\mathbf{r}_n) + e_a \psi_g^{a,\dagger} \psi_g^a(\mathbf{r}_n) \right], \tag{7}$$

and (4) becomes

$$H_{\text{int}} = -\sum_{ni} \left( p_i^+(\mathbf{r_n}) \, \mathbf{e}_i \cdot \mathbf{E}(\mathbf{r_n}) + \text{H.C.} \right), \tag{8}$$

with

$$p_i^+(\mathbf{r_n}) = d \sum_{ab} \int d\mathbf{r} \, \varphi_a^*(\mathbf{r})\varphi_b(\mathbf{r}) \, \psi_{e_i}^{a,\dagger}(\mathbf{r}_n)\psi_g^b(\mathbf{r}_n). \tag{9}$$

Equation (2), (7) and (8) describe the dynamics of the atomic array. Initially, we prepare all atoms in the $s$-wave internal ground state $|g\rangle$ with motional ground state $\varphi_0(\mathbf{r})$. The number of atoms in the excited states are suppressed due to the violation of energy conservation. We further add an external probe light, at fixed frequency $\omega$, which is near-resonant with $\delta \equiv \omega - \omega_0 \ll \omega, \omega_0$. The electric field reads $\mathbf{E}_0(\mathbf{r}) = \mathbf{E}_0 e^{i\mathbf{k}\cdot\mathbf{r}}$ with $c|\mathbf{k}| = \omega$. We take $c = 1$ from now on for conciseness.

This probe corresponds to the incident light in the scattering experiment. Its coupling to atoms reads

$$\delta H = -\sum_{ni} \left( p_i^+(\mathbf{r_n}) e^{-i\omega t} \mathbf{e}_i \cdot \mathbf{E}_0(\mathbf{r}_n) + \text{H.C.} \right). \tag{10}$$

We assume the field strength $\mathbf{E}_0$ is weak and the response can be analyzed using the linear response theory. The total electric field including the incident light and the scattered light then reads

$$\mathbf{E}_{\text{tot}}(\omega, \mathbf{r}) = \mathbf{E}_0(\mathbf{r}) + \langle \mathbf{E}(\omega, \mathbf{r}) \rangle. \tag{11}$$

Far from the atomic array, when only a single diffraction order exists, we expect

$$\mathbf{E}_{\text{tot}}(\omega, \mathbf{r}) = \left( \mathbf{1} e^{ik_z z} + \mathbf{S}(\omega, \mathbf{k}_\parallel) e^{ik_z |z|} \right) \cdot \mathbf{E}_0 e^{i\mathbf{k}_\parallel \cdot \mathbf{r}_\parallel}, \tag{12}$$

and $\mathbf{S}(\omega, \mathbf{k}_\parallel)$ is the corresponding $\mathbf{S}$ matrix.

## 3  Diagrammatic Expansion

The contribution to the scattered light $\langle \mathbf{E}(\omega, \mathbf{r}) \rangle$ can be efficiently organized using path-integral formulism. In particular, we work on the Keldysh contour [38], which contains a forwardly evolving branch and a backwardly evolving branch, corresponding to $e^{-iHt}$ and $e^{iHt}$ in the Heisenberg evolution. It is one of standard techniques for analyzing quantum many-body dynamics and disorder systems.

The expectation of fluctuation field becomes non-zero due to the coupling to atoms. Diagrammatically, we have

$$\langle \mathbf{E}(\omega, \mathbf{r}) \rangle = \overset{\mathbf{E}(\omega, \mathbf{r}) \qquad\qquad \mathbf{p}^-(\omega, \mathbf{r}_n)}{\sim\!\sim\!\sim\!\sim\!\sim\!\bullet}$$
$$= -\sum_n \mathbf{G}_{\text{R}}^{\mathbf{E}}(\omega, \mathbf{r} - \mathbf{r}_n) \cdot \langle \mathbf{p}^-(\omega, \mathbf{r}_n) \rangle. \tag{13}$$

Here we use the wavy line to represent the propagation of photons. $\mathbf{G}_{\text{R}}^{\mathbf{E}}$ is the retarded Green's function matrix of $\mathbf{E}$ in free space defined as

$$\mathbf{G}_{\text{R}}^{\mathbf{E}}(t, \mathbf{r}) \equiv -i\theta(t) \langle [E(t, \mathbf{r}), E(0, \mathbf{0})] \rangle_{d=0}. \tag{14}$$

In frequency and momentum space, we have

$$\tilde{\mathbf{G}}_{\text{R}}^{\mathbf{E}}(\omega, \mathbf{k}) = (\epsilon_0 \mathbf{1} + \epsilon_0 \mathbf{k} \times \mathbf{k} \times /\omega^2)^{-1} = -\frac{\omega^2}{\epsilon_0} \tilde{\mathbf{G}}(\omega, \mathbf{k}). \tag{15}$$

Here $\tilde{\mathbf{G}}(\omega, \mathbf{k})$ is the standard dyadic Green's function [16, 39]. Note that we have added an additional tilde for the Green's function of photons in momentum space to avoid possible confusion. The local dipole moment $\mathbf{p}^-$ is related to the incident light $\mathbf{E}_0$ by the Kubo formula [40]

$$\langle \mathbf{p}^-(\omega, \mathbf{r_n}) \rangle = \overset{\mathbf{p}^-(\omega, \mathbf{r}_n) \qquad\qquad \mathbf{p}^+(-\omega, \mathbf{r}_m)}{=\!=\!=\!=\!=}$$
$$= -\sum_m \mathbf{G}_{\text{R}}^{\mathbf{p}}(\omega, \mathbf{r}_{nm}) \cdot \mathbf{E}_0(\omega, \mathbf{r}_m). \tag{16}$$

We use the double solid line for the retarded Green's function for dipole momentums $\mathbf{G}_{\text{R}}^{\mathbf{p}}(\omega, \mathbf{r})$ and $\mathbf{r}_{nm} \equiv \mathbf{r}_n - \mathbf{r}_m$. This is consistent with the semi-classical analysis [16]. The remaining task is to derive approximate formula for $\mathbf{G}_{\text{R}}^{\mathbf{p}}(\omega, \mathbf{r})$, which includes renormalization due to the coupling with photons.

In this work, we take diagrams with single excitation which conserves the total energy. We first consider the correction of the excited state Green's function $G_R^e(\omega, \mathbf{r}, \mathbf{r}')$ by emission and absorption of photons. As we will see, since the wave function for ground-state atoms is localized, only $G_R^e(\omega, \mathbf{r}, \mathbf{r}')$ with $\mathbf{r} \approx \mathbf{r} \approx \mathbf{r}_n$ contributes to the light scattering. Here we assume the potential for the excited state is also deep, although the trapping frequency may differs from those of the ground state. As a result, we approximate the bare Green's function near $\mathbf{r}_n = \mathbf{0}$ as

$$G_R^{0,e}(\omega, \mathbf{r}, \mathbf{r}') \approx \sum_a \frac{\varphi_a'(\mathbf{r})\varphi_a'(\mathbf{r}')^*}{\omega - \omega_0 - e_a' + i\varepsilon}. \tag{17}$$

The Schwinger-Dyson equation reads $(G_R^e)^{-1} = (G_R^{0,e})^{-1} - \Sigma_R^e$, with the self-energy $\Sigma_R^e$

$$\Sigma_R^e(\omega, \mathbf{r}, \mathbf{r}') = \underset{\mathbf{E} \qquad e}{\overbrace{\longleftarrow \!\!\! \sim\!\!\!\sim\!\!\!\sim \!\!\! \bullet \longleftarrow}}$$

$$\approx -\frac{\omega^2 d^2}{\epsilon_0} G_{ii}(\omega, \mathbf{0}) \sum_a (1 - n_a)\varphi_a(\mathbf{r})\varphi_a^*(\mathbf{r}'). \tag{18}$$

Here $n_{a\geq 1} = 0$ and $n_0 = n$ is equal to the filling fraction. The appearance of $G_{ii}(\omega, \mathbf{0}) = \mathbf{e}_i \cdot \mathbf{G}(\omega, \mathbf{0}) \cdot \mathbf{e}_i$ owes to the approximation in (8) by using $\mathbf{E}(\mathbf{r}_n)$ instead of $\mathbf{E}(\mathbf{r})$. The real-part of $\mathbf{G}(\omega, \mathbf{0})$ contributes to the lamb shift, which can be absorbed in the definition of $\omega_0$. As a result, we only keep the imaginary part $\mathbf{G}(\omega, \mathbf{0}) = i\omega/6\pi$. We also assume $\delta, e_a, e_a' \ll \omega$, and the resonance frequency $\omega$ is much larger than the loop frequency, which is an analogy of the Markovian approximation in the master equation [16]. The natural linewidth of a transition with frequency $\omega$ is known to be $\gamma = \omega_0^3 d^2/3\pi\epsilon_0$. This leads to

$$\Sigma_R^e(\omega, \mathbf{r}, \mathbf{r}') \approx -\frac{i\gamma}{2}\left[\delta(\mathbf{r} - \mathbf{r}') - n\varphi_0(\mathbf{r})\varphi_0^*(\mathbf{r}')\right], \tag{19}$$

where we have used the completeness of local wave functions $\sum_a \varphi_a(\mathbf{r})\varphi_a^*(\mathbf{r}') = \delta(\mathbf{r} - \mathbf{r}')$.

Having obtained the dressed Green's function, we turn to the calculation of $\mathbf{G}_R^p(\omega, \mathbf{r})$. Motivated by the standard Random Phase Approximation (RPA) in interacting fermions [41], we consider the diagrams

$$\mathbf{G}_R^p = \underset{g}{\overset{e,i}{\langle\delta_{mn}\rangle}} + \underset{g}{\overset{e,i}{\langle\mathbf{r}_n\rangle}} \!\!\sim\!\!\! \underset{g}{\overset{e,i}{\langle\mathbf{r}_m\rangle}} \ldots \tag{20}$$

The thick solid line represents the normalized Green's function $G_R^e$. The first bubble diagram, which is an elementary building block, is given by

$$i\Pi_R(\omega) = \frac{d^2}{2}\int \frac{d\tilde{\omega}}{2\pi}d\mathbf{r}'d\mathbf{r}\, G_R^e(\tilde{\omega} + \omega, \mathbf{r}, \mathbf{r}')G_K^g(\tilde{\omega}, \mathbf{r}', \mathbf{r}) + G_K^e(\tilde{\omega} + \omega, \mathbf{r}, \mathbf{r}')G_A^g(\tilde{\omega}, \mathbf{r}', \mathbf{r}). \tag{21}$$

Here $G_A^\eta$ is the advanced Green's function. $G_K^\eta = G_R^\eta \circ F_\eta - F_\eta \circ G_A^\eta$ is the Keldysh Green's function, and $F_\eta = (1 - 2n_\eta)$ is the quantum distribution function [38]. It can be further simplified as

$$\Pi_R(\omega) = d^2 n \int d\mathbf{r}'d\mathbf{r}\, G_R^e(\omega + e_0, \mathbf{r}, \mathbf{r}')\varphi_0(\mathbf{r})^*\varphi_0(\mathbf{r}'). \tag{22}$$

Summing over the diagrams with multiple bubbles, in momentum space, we have

$$i\mathbf{G}_R^p(\omega, \mathbf{k}_\parallel) = i\Pi_R(\omega) - i\Pi_R(\omega)i\tilde{\mathcal{G}}_R^{\mathbf{E}}(\omega, \mathbf{k}_\parallel)i\Pi_R(\omega) + \ldots$$

$$= \frac{i}{\Pi_R(\omega)^{-1}\mathbf{1} - \tilde{\mathcal{G}}_R^{\mathbf{E}}(\omega, \mathbf{k}_\parallel)}. \tag{23}$$

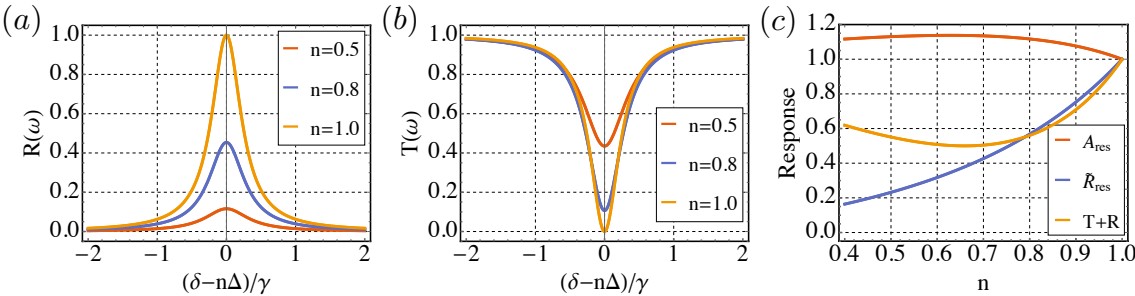

Figure 2: Numerical result for the fractional-filling effect with normal incident light with $\omega a = 2\pi \times 0.68$. Here we take $V_e(\mathbf{r}) = V_g(\mathbf{r})$. (a). The reflection coefficient $R(\omega)$ as a function of detuning $\delta - n\Delta$ for different filling fraction $n$. (b). The transmission coefficient $T(\omega)$ as a function of detuning $\delta - n\Delta$ for different filling number $n$. (c). The filling-normalized absorptance $A$ and reflectance $\tilde{R}$, together with $T + R$ at cooperative resonance $\delta = n\Delta$, as a function of filling fraction $n$.

Since the summation in (16) is descrete, the Fourier transform is defined as

$$\tilde{\mathcal{G}}_R^{\mathbf{E}}(\omega, \mathbf{k}_\parallel) = \sum_n G_R^{\mathbf{E}}(\omega, \mathbf{k}_\parallel) e^{-i\mathbf{k}_\parallel \cdot \mathbf{r}_n}. \tag{24}$$

In particular, the denominator of (23) is a generalization of the corresponding result under non-Hermitian Hamiltonians, which is $\omega \mathbf{1} - \mathcal{H}_{\text{eff}}$. As we will see later, (21) takes such a form only for unit filling and $V_e(r) = V_g(r)$. This implies the breakdown of non-Hermitian Hamiltonian description for general setups.

Then, using the relation (15), we obtain the relation between $\langle \mathbf{p}^- \rangle$ and $\mathbf{E}_0$ in momentum space as

$$
\begin{aligned}
\langle \mathbf{p}^-(\omega, \mathbf{k}_\parallel) \rangle &= \boldsymbol{\alpha}(\omega, \mathbf{k}_\parallel) \cdot \mathbf{E}_0(\mathbf{k}_\parallel), \\
\boldsymbol{\alpha}(\omega, \mathbf{k}_\parallel)^{-1} &= -\mathbf{1}/\Pi_R(\omega) + \tilde{\mathcal{G}}_R^{\mathbf{E}}(\omega, \mathbf{k}_\parallel),
\end{aligned}
\tag{25}
$$

Finally, for a single diffraction order, using (13), $\boldsymbol{\alpha}$ is related to the $S$ matrix as [16]

$$\mathbf{S}(\omega, \mathbf{k}_\parallel) = \frac{i\omega^2}{2a^2 \epsilon_0 k_z} \mathcal{P}(\omega, \mathbf{k}_\parallel) \cdot \boldsymbol{\alpha}(\omega, \mathbf{k}_\parallel). \tag{26}$$

Here $\mathcal{P}_{ij}(\omega, \mathbf{k}_\parallel) = \delta_{ij} - \xi_{ij} \frac{k_i k_j}{\omega^2}$. $\xi_{ij} = -1$ if only one of $(i, j)$ is in $z$ direction, and at the same time $z < 0$. In other cases, $\xi_{ij} = 1$.

Equation (18), (22), (25), and (26) together determine the cooperative optical response of the atomic array. In the next sections, we further focus on two different simple setups to study the cooperative resonances in the atomic array below unit filling and the effect of trapping using the formulism developed above. Both of these effects have been observed in the recent experimental realization of the atomic array [1].

## 4 Fractional Filling

We begin with analyzing the effect of fractional filling $n < 1$. Here we take the special case $V_e(\mathbf{r}) = V_g(\mathbf{r})$, which is valid when the optical lattice is formed by a light with the magic wavelength [42]. This leads to $\varphi_a'(\mathbf{r}) = \varphi_a(\mathbf{r})$, $e_a' = e_a$, and (9) becomes diagonal in $a$. As a result, the transition of internal state does not couple to the motional degree of freedom. Since the initial state only contains fermions in the motional ground state $g = 0$, motional excited states are never

occupied. Consequently, we only keep the $a = 0$ part of the Green's function. Projecting into the corresponding subspace, we have

$$\Sigma_R^e(\omega, \mathbf{r}, \mathbf{r}') \approx -\frac{i\gamma}{2}(1-n)\varphi_0(\mathbf{r})\varphi_0^*(\mathbf{r}'),$$

$$G_R^e(\omega, \mathbf{r}, \mathbf{r}') \approx \frac{\varphi_0(\mathbf{r})\varphi_0(\mathbf{r}')^*}{\omega - \omega_0 - e_0 + i\frac{\gamma(1-n)}{2}}. \tag{27}$$

The bubble $i\Pi_R(\omega)$ is also simplified since the integral over $\mathbf{r}$ and $\mathbf{r}'$ is now trivial. We arrive at

$$\Pi_R(\omega) = \frac{d^2 n}{\omega - \omega_0 + i\frac{\gamma(1-n)}{2}} = \frac{d^2 n}{\delta + i\frac{\gamma(1-n)}{2}}. \tag{28}$$

Now we focus on the normal incident case as in experiment [1]. In this case, we have $k_x = k_y = 0$, $k_z = \omega$. Moreover, $\mathbf{E}_0$ lies in the $x$-$y$ plane, and we can take $\mathcal{P} = \mathbf{1}$ and $\alpha = \alpha\mathbf{1}$ as scalars. Following the convention [16], we define

$$\mathbf{\Delta}(\mathbf{k}_\parallel) = -\frac{3\pi\gamma}{\omega}\sum_{n\neq 0}\mathrm{Re}\, \mathbf{G}(\omega, \mathbf{r}_n)e^{-i\mathbf{r_n}\cdot\mathbf{k}_\parallel},$$

$$\mathbf{\Gamma}(\mathbf{k}_\parallel) = \frac{6\pi\gamma}{\omega}\sum_{n\neq 0}\mathrm{Im}\, \mathbf{G}(\omega, \mathbf{r}_n)e^{-i\mathbf{r_n}\cdot\mathbf{k}_\parallel}, \tag{29}$$

which also become scalars $\Delta$ and $\Gamma$ in the $x$-$y$ plane for normal incident light. In particular, it is known that $\Gamma + \gamma = \gamma\frac{3\pi}{a^2\omega^2}$ [16]. Using these definition, we have

$$\alpha = -\frac{6\pi\epsilon_0}{\omega_0^3}\frac{n\gamma/2}{\delta - n\Delta + i(\gamma + n\Gamma)/2}, \qquad S = -\frac{in(\gamma + \Gamma)/2}{\delta - n\Delta + i(\gamma + n\Gamma)/2}. \tag{30}$$

The cooperative linewidth becomes $\gamma + n\Gamma$. For $n < 1$, we find $|S| < 1$ even at the resonance and the mirror becomes imperfect. The transmission coefficient $T = |1 + S|^2$ and reflection coefficient $R = |S|^2$ are found to be

$$T = \frac{(\delta - n\Delta)^2 + (1-n)^2\gamma^2/4}{(\delta - n\Delta)^2 + (\gamma + n\Gamma)^2/4}, \qquad R = \frac{n^2(\gamma + \Gamma)^2/4}{(\delta - n\Delta)^2 + (\gamma + n\Gamma)^2/4}. \tag{31}$$

The filling-normalized absorptance $A = (1-T)/n$ and reflectance $\tilde{R} = R/n$ then reads

$$A = \frac{(n\Gamma + (2-n)\gamma)(\gamma + \Gamma)/4}{(\delta - n\Delta)^2 + (\gamma + n\Gamma)^2/4}, \qquad \tilde{R} = \frac{n(\gamma + \Gamma)^2/4}{(\delta - n\Delta)^2 + (\gamma + n\Gamma)^2/4}. \tag{32}$$

We plot the numerical result for $\omega a = 2\pi \times 0.68$ as in the experiment [1] for various $n$ in Figure 2, where we have $\Delta/\gamma \approx 0.18$ and $\Gamma/\gamma \approx -0.48$. Several comments are as follows

1. All above results reduces to the semi-classical results using spin operators when $n = 1$, where the frequency shift is $\Delta$ and the linewidth becomes $\gamma + \Gamma$. On the other hand, for $n \to 0$, we get back to the single-atom response with natural linewidth $\gamma$. For general $n$, the frequency shift $n\Delta$ and linewidth $\gamma + n\Gamma$ is linear in $n$, consistent with the experimental observation and numerical simulation in [1]. For $\Gamma < 0$, this corresponds to a suppression of the subradiance. As we will see in the next section, the linear dependence is universal and also valid for $V_e(\mathbf{r}) \neq V_g(\mathbf{r})$.

2. As observed in the experiment [1], generally, we have $T + R < 1$. This is due to the fact that the self-energy of the excited state (18) contains the contribution of spontaneous emission of photons in arbitrary directions with random phases, which can not be observed by averaged $\mathbf{E}_{\mathrm{tot}}$. However, the corresponding contribution exists if we measure energy density of electromagnetic field $\langle E^2(\mathbf{r})\rangle$.

3. The filling-normalized absorptance $A$ show a weak dependence of $n$, while $\tilde{R}$ vanishes as $n \to 0$, consistent with the experimental observation and numerical simulation in [1].

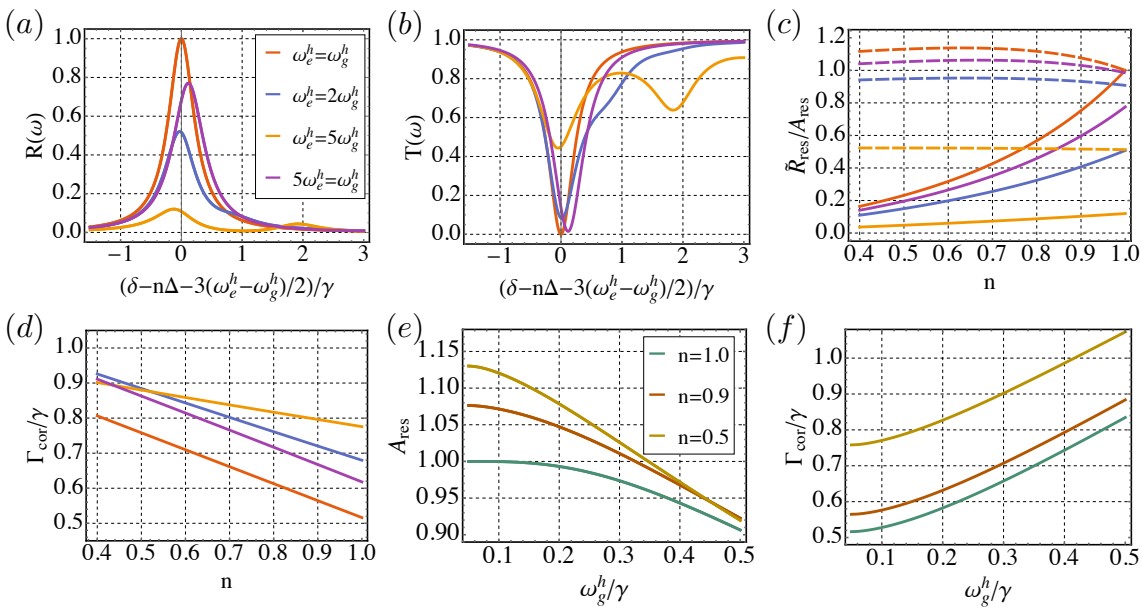

Figure 3: Numerical result for the trapping effect with normal incident light with $\omega a = 2\pi \times 0.68$. We fix $\omega_g^h = \gamma/4$ in (a-d) and $\omega_e^h = \gamma/20$ in (e-f). (a). The reflection coefficient $R(\omega)$ as a function of detuning for $n = 1$ with different $\omega_e^h/\omega_g^h$. (b). The transmission coefficient $T(\omega)$ as a function of detuning for $n = 1$ with different $\omega_e^h/\omega_g^h$. (c). The fitted $A_{\mathrm{res}}$ and $\tilde{R}_{\mathrm{res}}$ as a function of $n$ for different $\omega_e^h/\omega_g^h$. Here the dashed lines corresponds to $A_{\mathrm{res}}$. (d). The fitted linewidth $\Gamma_{\mathrm{cor}}$ as a function of $n$ for different $\omega_e^h/\omega_g^h$. (e). The fitted $A_{\mathrm{res}}$ as a function of $\omega_g^h$ for different $n$. (f). The fitted linewidth $\Gamma_{\mathrm{cor}}$ as a function of $\omega_g^h$ for different $n$.

## 5 Trapping Mismatch

In this section, we discuss the effect of having $V_e(\mathbf{r}) \neq V_g(\mathbf{r})$. We further expand the potential of near the minimum of each site and use the approximation of 3D isotropic harmonic potential. Ground-state atoms $|g\rangle$ and excited-state atoms $|e_i\rangle$ have a trapping frequency $\omega_g^h$ and $\omega_e^h$ correspondingly. The motional ground state wave function $\varphi_0(\mathbf{r})$ reads

$$\varphi_0(\mathbf{r}) = \left(\frac{\omega_g^h}{\pi}\right)^{\frac{3}{4}} e^{-\frac{\omega_g^h r^2}{2}}. \tag{33}$$

The dressed Green's function $G_{\mathrm{R}}^e$ can be further simplified by separating the contribution from the delta function and $\varphi_0$ (19). We have

$$\varphi_0^* \circ G_{\mathrm{R}}^e \circ \varphi_0 = \sum_a \frac{(\varphi_0 \circ \varphi_a')^* \varphi_a' \circ \varphi_0}{\delta + e_0 - e_a' + \frac{i\gamma}{2}} + \frac{i\gamma n}{2}\left(\sum_a \frac{(\varphi_0 \circ \varphi_a')^* \varphi_a' \circ \varphi_0}{\delta + e_0 - e_a' + \frac{i\gamma}{2}}\right)^2 + \dots \tag{34}$$

Here we use $\circ$ to represent the integral over spatial dimension for conciseness. If we define summing up the geometric series, we find

$$\Pi_{\mathrm{R}}(\omega) = \frac{d^2 n}{\pi(\omega)^{-1} - i\frac{\gamma n}{2}}, \qquad \pi(\omega) \equiv \sum_a \frac{(\varphi_0 \circ \varphi_a')^* \varphi_a' \circ \varphi_0}{\delta + e_0 - e_a' + \frac{i\gamma}{2}}. \tag{35}$$

The analytical expression for $\pi(\omega)$ is presented in Appendix A. It contains multiple resonances near $\delta = (3/2 + 2n)\omega_e^h - 3\omega_g^h/2$, broadened by the natural lifetime $\gamma$ of the excited state. For

$\omega_e^h \gtrsim \gamma$, this leads to different peaks in the spectral $-\text{Im}\pi(\omega)/\pi$. For $\omega_e^h \lesssim \gamma$, different resonances merges, and only a singe peak exists.

Again, we focus on the normal incident case. The $S$ matrix can be obtained as

$$\alpha = -\frac{6\pi\epsilon_0}{\omega_0^3} \frac{n\gamma/2}{\pi^{-1} - n\Delta + in\Gamma/2}, \qquad S = -\frac{in(\gamma + \Gamma)/2}{\pi^{-1} - n\Delta + in\Gamma/2}. \tag{36}$$

It is straightforward to check that (36) reduces to (30) when $\omega_1 = \omega_2$. Since $\pi(\omega)$ is independent of $n$, the cooperative linewidth is still linear in filling fraction $n$.

The parameters in the experiment [1] corresponds to $\omega_g^h < \gamma$, but at the same order $\sim$ MHz. We plot our results (36) for different $\omega_e^h/\gamma$, $\omega_g^h/\gamma$ and $n$ in Figure 3. We first fix $\omega_g^h/\gamma = 1/4$ and study the effect of $\omega_e^h \neq \omega_g^h$. As shown in (a) and (b), either $\omega_e^h > \omega_g^h$ or $\omega_g^h > \omega_e^h$, the atomic mirror becomes imperfect with max $R < 1$ and min $T > 0$. For $\omega_e^h \gtrsim \gamma$, we see multi-peak structures at energy $2n\omega_e^h$, where the transition from $|g\rangle$ to $|e_i\rangle$ is accompanied with the excitation of motional degree of freedom. For small $\omega_e^h \lesssim \gamma$, both $R(\omega)$ and $T(\omega)$ show a single Lorentzian peak.

Motivated by the experimental result, we study the the cooperative linewidth of the atomic array by fitting the numerical result for $R(\omega)$ near $\delta = n\Delta + 3(\omega_e^h - \omega_g^h)/2$ as

$$R(\omega) = \frac{R_{\text{res}}\Gamma_{\text{cor}}^2/4}{(\delta - n\Delta - 3(\omega_e^h - \omega_g^h)/2 - \delta_0)^2 + \Gamma_{\text{cor}}^2/4}, \tag{37}$$

and define $T_{\text{res}} = T(n\Delta + 3(\omega_e^h - \omega_g^h)/2 + \delta_0)$. $\tilde{R}_{\text{res}}$ and $A_{\text{res}}$ can then be computed correspondingly using $R_{\text{res}}$ and $T_{\text{res}}$. The numerical results in (c-d) show $\tilde{R}_{\text{res}}$ and $A_{\text{res}}$ also decreases when $\omega_e^h \neq \omega_g^h$, and $\Gamma_{\text{cor}}$ is linear in $n$. However, the linewidth for $\omega_e^h > \gamma$ receives corrections from multi-peaks. We then study $A_{\text{res}}$ and $\Gamma_{\text{cor}}$ as a function of $\omega_g^h$. We fix a small $\omega_e^h = \gamma/20$, as an analogy of the anti-trapped excited state in experiment [1], and tune $\omega_g^h$. We find for small $\omega_g^h$, the decrease in $A_{\text{res}}$ and the increase of the decay rate show quadratic dependence, while for large $\omega_g^h$, the dependence becomes linear. This is a close analogy of the experimental observation in [1].

# 6 Summary and Overlook

In this work, we study quantum atomic arrays using a microscopic model with atoms in optical lattices. We take a diagrammatic approach with PRA-like diagrams and obtain concise results for transmission and reflection coefficients. We find both fractional filling and trapping mismatch can result in the imperfectness of mirrors. For the fractional filling, we derive the cooperative lifetime and resonant frequency, which show linear dependence with the filling fraction $n$. We also find multiple peaks exist when the local trapping frequency of the excited state $\omega_e^g \sim \gamma$, and study the trapping frequency effects on the cooperative lifetime.

Our results can be tested in the experimental platforms similar to that in [1]. Recently, there are also experimental studies on the Pauli blocking of light scattering in degenerate fermions [43,44]. The diagrammatic approach developed here can also be applied to study the optical response of degenerate fermion gases.

# Acknowledgments

We especially thank Yu Chen and Jianwen Jie for helpful discussions. P.Z. acknowledges support from the Walter Burke Institute for Theoretical Physics at Caltech.

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

# A The Analytical Formula for $\pi(\omega)$

In this Appendix, we present detailed derivation of the analytical formula for $\pi(\omega)$. We trick is to use the transformation to the time domain

$$
\begin{aligned}
\pi(\omega) &= \sum_a \int d\mathbf{r} d\mathbf{r}' \, \varphi_0(\mathbf{r})^* \varphi_a'(\mathbf{r}) \frac{1}{\delta + e_0 - e_a' + \frac{i\gamma}{2}} \varphi_a'(\mathbf{r}')^* \varphi_0(\mathbf{r}') \\
&= -\sum_a \int d\mathbf{r} d\mathbf{r}' \int_0^\infty d\tau \, \varphi_0(\mathbf{r})^* \varphi_a'(\mathbf{r}) e^{(\delta + e_0 - e_a' + \frac{i\gamma}{2})\tau} \varphi_a'(\mathbf{r}')^* \varphi_0(\mathbf{r}') \\
&= -\int d\mathbf{r} d\mathbf{r}' \int_0^\infty d\tau \, e^{(\delta + e_0 + \frac{i\gamma}{2})\tau} \varphi_0(\mathbf{r})^* K_{\omega_e^h}(\tau, \mathbf{r}, \mathbf{r}') \varphi_0(\mathbf{r}').
\end{aligned}
\tag{38}
$$

Here we have assumed the integral over $\tau$ is convergent by restricting the $\delta + e_0 < 3\omega_e^h/2$. After the integration, analytical continuation can be applied to release this restriction. Here $K_{\omega_e^h}(\tau, \mathbf{r}, \mathbf{r}')$ is the imaginary time Green's function in a harmonic trap with trapping frequency $\omega_e^h$. We have

$$
K_{\omega_e^h}(\tau, \mathbf{r}, \mathbf{r}') = \left( \frac{\omega_e^h}{2\pi \sinh \omega_e^h \tau} \right)^{\frac{3}{2}} \exp\left( -\frac{\omega_e^h}{2} \left[ (r^2 + r'^2) \coth \omega_e^h \tau - \frac{2\mathbf{r} \cdot \mathbf{r}'}{\sinh \omega_e^h \tau} \right] \right).
\tag{39}
$$

The integral over $\mathbf{r}$ and $\mathbf{r}'$ can be carried out first. We find

$$
\pi(\omega) = -2\sqrt{2} \int_0^\infty d\tau \, e^{a_0 \tau} \left( \frac{\omega_e^h \omega_g^h}{2\omega_e^h \omega_g^h \cosh \omega_e^h \tau + [(\omega_e^h)^2 + (\omega_g^h)^2] \sinh \omega_e^h \tau} \right)^{\frac{3}{2}}.
\tag{40}
$$

Here we have defined $a_0 = \left( \delta + \frac{3\omega_g^h}{2} + \frac{i\gamma}{2} \right)$ for conciseness. Then the integral over $\tau$ gives

$$
\pi(\omega) = \frac{\sqrt{2}}{\omega_e^h} \left( \frac{2\omega_e^h \omega_g^h}{(\omega_e^h + \omega_g^h)^2} \right)^{\frac{3}{2}} \frac{q^{-p-1} \left( (-2p(q-1) - q + 2) B_q \left( p+1, \frac{1}{2} \right) - (2p+3) B_q \left( p+1, \frac{3}{2} \right) \right)}{1-q},
\tag{41}
$$

where $p \equiv -\frac{2a_0 + \omega_e^h}{4\omega_e^h}$ and $q \equiv \frac{(\omega_e^h + \omega_g^h)^2}{(\omega_e^h - \omega_g^h)^2}$. $B_z(a,b)$ is the incomplete beta function defined as $B_z(a,b) = \int_0^z t^{a-1}(1-t)^{b-1} dt$. For $\omega_e^h = \omega_g^h$, one can check that above result can be simplified as $\pi(\omega)^{-1} = a_0 - 3\omega_e^h/2$.