# Peer review of "Quantum Atomic Arrays: Fractional Filling and Trapping"

_SciPost Physics_

## Round 1 · Referee Report · Anonymous (Referee 1) · 2021-9-17

Strengths

  1. The work presented simple analytical results qualitatively reproducing experimental trends.

  2. It provides one of the first studies of trap mismatch between atomic levels

Weaknesses

  1. The diagrammatic approach contains a mistake in the RPA series of (20): the insertion of the self-energy (18) is not allowed. This mistake leads to a wrong (1-n) factor multiplying \gamma in the denominator of (28)

  2. The above unphysical factor is neglected in all the following equations without any explanation. On the other hand, the self-energy (18), which leads to this unphysical factor, is used as an explanation for the experimental observation that T+R<1.

  3. One of the two main results, namely the linear scaling with n of several observables, has been demonstrated elsewhere [PRA 102, 033720 (2020), Eq.(39)] with the same diagrammatic methods. The corresponding literature is not cited.

Report

I would like to recommend this paper for publication since it nicely demonstrates how such non-equilibrium diagrammatic methods allow for simple analytical results with direct experimental relevance for situations like Ref.[1]. However, I have two main reservations.

The first is technical. The very method the Authors promote here is used wrongly, even though only the simplest leading order bubble-diagram is considered. When mapping the spin variable using (bosonic or fermionic) creation and annihilation operators, one further needs to enforce that the atom is either in the excited or in the ground state, or in a superposition. As shown in [PRA 102, 033720 (2020)], within this diagrammatic description this needs to be enforced via additional Feynman rules, according to which certain diagrams need to be discarded. The Authors do not discuss this at all and keep one of those unphysical diagrams, namely the one obtained by dressing the excited-state propagator in (20) with the self-energy (18). These issues are discussed in [PRA 102, 033720 (2020),PRL 125, 133604 (2020)] also at the non-linear (many-body) level. The unphysical diagram leads to the unphysical factor (1-n) multiplying \gamma in eq.(28).
What is strange is that afterwards, without any comment, the Authors neglect this unphysical factor so that it does not affect any of the results presented. While neglecting this factor is essential to match the results with the previous literature for n=1, it struck me as odd that this is done without mention.

The second reservation is about the content, as I am not sure there is enough original results here. This is due to the fact that one of the main findings, namely the linear scaling with n of the RPA polarisability, has been obtained in the above mentioned work [PRA 102, 033720 (2020), Eq.(39), where 1-nV there corresponds to n here] with the same diagrammatic methods. Thus the only novel analysis is the study of the trap mismatch. This is actually interesting and demonstrated the power of the methods used. I feel that this part should be somewhat extended and at the same time the previous literature properly acknowledged in order for this work to be substantial enough to deserve publication.

Requested changes

  1. Correct the mistake in the diagrammatic method (see above), provide a discussion of the issue, and refer to the relevant literature.

  2. Remove remark 2. on page 7 since it is based on the self-energy (18) which should not be included.

  3. Extend the part dealing with the trap mismatch. A part from exploring different parameter regimes, one further interesting extension could be the analysis of recoil, as done in the recent works [PRA 99, 013410 (2019); PRA 103, 043722 (2021)].

  4. Correct English language mistakes for example the title of the section "Summary and Overlook" should read "Summary and Outlook".

  • validity: low
  • significance: good
  • originality: low
  • clarity: ok
  • formatting: reasonable
  • grammar: below threshold

---

## Round 1 · Referee Report · Anonymous (Referee 2) · 2021-10-13

Strengths

1-Interesting study of the effect of fractional filling and trapping mismatch in an atomic array on the scattering of an external light field.

Weaknesses

1-Poorly written manuscript with an enormous amount of undefined quantities and typos.

2-Hardly reproducible because of the very weak explanation and the many undefined quantities.

Report

Pengfei Zhang employs a Green's function formalism to study the scattering of an external light field interacting with a dipolar atomic array. The author describes the dipolar atoms to be tightly confined in an optical lattice that experience an external monochromatic probe field. Here, the author studies the response of the dipoles and is able to determine the scattered light with a diagramatic approach. The presented methods allow to include trapping mismatch (between the ground and excited states of the dipole) and fractional filling of lattice sites and study their effects onto the scattered light field. The latter is done in sections 4 and 5 of the manuscript. Here, it is shown that reducing the filling leads to a reduction of the reflection coefficient and to an increase of the transmission coefficient. The author also studies the modification of these coefficients when the excited and ground states experience different trapping frequencies.

I believe that the topic of this work is timely and the results are useful for the physics community working with atomic arrays. However, this alone is not sufficient to be accepted in SciPost Physics and I do not see any of the four "expectations" fulfilled. Moreover, my opinion is that this paper must have been written in a great hurry since it includes many typos and many quantities are not defined or explained in the manuscript. As a result it is hard to reproduce the presented results and to follow the line of thought in the calculations. I want to show this problem exemplarily on the model section 2 (page 2):

(I) How can I assume that all p-wave excited states see the same optical lattice (Eq.(3))?
(II) Before Eq. (4) in the anticommutation relation it should be $\psi_b^{\dagger}$ I suppose.
(III) The author uses "a" for the lattice constant. The variable "a" is also used in the fermionic annihilation and creation operators (I guess that this is the band [undefined]?)
(IV) Those annihilation and creation operators are decomposed in local operators (Eq.(6)) with local wave-functions; I guess those should be the Wannier functions?
(V) Also in Eq. (6) the summation runs over $n$ that are probably the lattice sites, so actually $n_1$ and $n_2$?
(VI) In Eq. (9) the overlap integral is calculated between $\varphi_a$ and $\varphi_b$; that should be a $\delta_{ab}$. However, I guess what the author wanted to write was the overlap between $\varphi_a'$ and $\varphi_b$ that is not a Kronecker-delta (see also the comment in the beginning of section 4).

These things can all be found on one page out of nine pages and this sloppiness is found on almost every page of the manuscript. Therefore, my opinion is that this manuscript should not be published in SciPost Physics nor anywhere else before the author has not significantly improved the explanations of the expressions as well as the presentation of all derived quantities. This is also the minimum requirement to assure that the presented results are correct.
  • validity: ok
  • significance: low
  • originality: ok
  • clarity: poor
  • formatting: below threshold
  • grammar: below threshold

---

## Editorial Decision

resubmitted